# Model-based Policy Optimization with Unsupervised Model Adaptation

**Jian Shen**[‡], **Han Zhao**[⋆*], **Weinan Zhang**[‡†] , **Yong Yu**[‡]
[‡]Shanghai Jiao Tong University, [⋆]D. E. Shaw & Co
{rockyshen, wnzhang, yyu}@apex.sjtu.edu.cn
han.zhao@cs.cmu.edu

## Abstract

Model-based reinforcement learning methods learn a dynamics model with real data sampled from the environment and leverage it to generate simulated data to derive an agent. However, due to the potential distribution mismatch between simulated data and real data, this could lead to degraded performance. Despite much effort being devoted to reducing this distribution mismatch, existing methods fail to solve it explicitly. In this paper, we investigate how to bridge the gap between real and simulated data due to inaccurate model estimation for better policy optimization. To begin with, we first derive a lower bound of the expected return, which naturally inspires a bound maximization algorithm by aligning the simulated and real data distributions. To this end, we propose a novel model-based reinforcement learning framework AMPO, which introduces unsupervised model adaptation to minimize the integral probability metric (IPM) between feature distributions from real and simulated data. Instantiating our framework with Wasserstein-1 distance gives a practical model-based approach. Empirically, our approach achieves state-of-the-art performance in terms of sample efficiency on a range of continuous control benchmark tasks.

## 1 Introduction

In recent years, model-free reinforcement learning (MFRL) has achieved tremendous success on a wide range of simulated domains, *e.g.*, video games [Mnih et al., 2015], complex robotic tasks [Haarnoja et al., 2018], just to name a few. However, model-free methods are notoriously data inefficient and often require a massive number of samples from the environment. In many high-stakes real-world applications, *e.g.*, autonomous driving, online education, etc., it is often expensive, or even infeasible, to collect such large-scale datasets. On the other hand, model-based reinforcement learning (MBRL), in contrast, is considered to be an appealing alternative that is able to substantially reduce sample complexity [Sun et al., 2018, Langlois et al., 2019].

At a colloquial level, model-based approaches build a predictive model of the environment dynamics and generate simulated rollouts from it to derive a policy [Janner et al., 2019, Luo et al., 2018, Kaiser et al., 2019] or a planner [Chua et al., 2018, Hafner et al., 2019]. However, the asymptotic performance of MBRL methods often lags behind their model-free counterparts, mainly due to the fact that the model learned from finite data can still be far away from the underlying dynamics of the environment. To be precise, even equipped with a high-capacity model, such model error still exists due to the potential distribution mismatch between the training and generating phases, *i.e.*, the state-action input distribution used to train the model is different from the one generated by the model [Talvitie, 2014]. Because of this gap, the learned model may give inaccurate predictions

---

[*]Work done while at Carnegie Mellon University. [†]Weinan Zhang is the corresponding author.

on simulated data and the errors can compound for multi-step rollouts [Asadi et al., 2018], which will be exploited by the follow-up policy optimization or planning procedure, leading to degraded performance.

In the literature, there is a fruitful line of works focusing on reducing the distribution mismatch problem by improving the approximation accuracy of model learning, or by designing careful strategies when using the model for simulation. For model learning, different architectures [Asadi et al., 2018, Asadi et al., 2019, Chua et al., 2018] and loss functions [Farahmand, 2018, Wu et al., 2019] have been proposed to mitigate overfitting or improve multi-step predictions so that the simulated data generated by the model are more like real. For model usage, delicate rollout schemes [Janner et al., 2019, Buckman et al., 2018, Nguyen et al., 2018, Xiao et al., 2019] have been adopted to exploit the model before the simulated data departure from the real distribution. Although these existing methods help alleviate the distribution mismatch, this problem still exists.

In this paper, we take a step further towards the goal of explicit mitigation of the distribution mismatch problem for better policy optimization in Dyna-style MBRL [Sutton, 1990]. To begin with, we derive a lower bound of the expected return in the real environment, which naturally inspires a bound maximization algorithm according to the theory of unsupervised domain adaptation. To this end, we propose a novel model-based framework, namely AMPO (Adaptation augmented Model-based Policy Optimization), by introducing a model adaptation procedure upon the existing MBPO [Janner et al., 2019] method. To be specific, model adaptation encourages the model to learn invariant feature representations by minimizing integral probability metric (IPM) between the feature distributions of real data and simulated data. By instantiating our framework with Wasserstein-1 distance [Villani, 2008], we obtain a practical method. We evaluate our method on challenging continuous control benchmark tasks, and the experimental results demonstrate that the proposed AMPO achieves better performance against state-of-the-art MBRL and MFRL methods in terms of sample efficiency.

## 2 Preliminaries

We first introduce the notation used throughout the paper and briefly discuss the problem setup of reinforcement learning and concepts related to integral probability metric.

**Reinforcement Learning**  A Markov decision process (MDP) is defined by the tuple $(\mathcal{S}, \mathcal{A}, T, r, \gamma)$, where $\mathcal{S}$ and $\mathcal{A}$ are the state and action spaces, respectively. Throughout the paper, we assume that the state space is continuous and compact. $\gamma \in (0, 1)$ is the discount factor. $T(s' \mid s, a)$ is the transition density of state $s'$ given action $a$ made under state $s$, and the reward function is denoted as $r(s, a)$. The goal of reinforcement learning (RL) is to find the optimal policy $\pi^*$ that maximizes the expected return (sum of discounted rewards), denoted by $\eta$:

$$\pi^* := \arg\max_\pi \eta[\pi] = \arg\max_\pi \mathbb{E}_\pi \left[ \sum_{t=0}^\infty \gamma^t r(s_t, a_t) \right], \tag{1}$$

where $s_{t+1} \sim T(s \mid s_t, a_t)$ and $a_t \sim \pi(a \mid s_t)$. In practice, the groundtruth transition $T$ is unknown and MBRL methods aim to construct a model $\hat{T}$ of the transition dynamics, using data collected from interaction with the MDP. Furthermore, different from several previous MBRL works [Chua et al., 2018, Luo et al., 2018], the reward function $r(s, a)$ is also unknown throughout the paper, and an agent needs to learn the reward function simultaneously.

For a policy $\pi$, we define the normalized occupancy measure, $\rho_{\hat{T}}^\pi(s, a)$ [Ho and Ermon, 2016], as the discounted distribution of the states and actions visited by the policy $\pi$ on the dynamics model $\hat{T}$: $\rho_{\hat{T}}^\pi(s, a) = (1 - \gamma) \cdot \pi(a \mid s) \sum_{t=0}^\infty \gamma^t P_{\hat{T}, t}^\pi(s)$, where $P_{\hat{T}, t}^\pi(s)$ denotes the density of state $s$ visited by $\pi$ under $\hat{T}$ at time step $t$. Similarly, $\rho_T^\pi(s, a)$ represents the discounted occupancy measure visited by $\pi$ under the real dynamics $T$. Using this definition, we can equivalently express the objective function as follows: $\eta[\pi] = \mathbb{E}_{\rho_T^\pi(s,a)}[r(s, a)] = \int \rho_T^\pi(s, a) r(s, a) \, \mathrm{d}s \, \mathrm{d}a$. To simplify the notation, we also define the normalized state visit distribution as $\nu_T^\pi(s) := (1 - \gamma) \sum_{t=0}^\infty \gamma^t P_{T, t}^\pi(s)$.

**Integral Probability Metric**  Integral probability metric (IPM) is a family of discrepancy measures between two distributions over the same space [Müller, 1997, Sriperumbudur et al., 2009].

Specifically, given two probability distributions $\mathbb{P}$ and $\mathbb{Q}$ over $\mathcal{X}$, the $\mathcal{F}$-IPM is defined as

$$d_{\mathcal{F}}(\mathbb{P}, \mathbb{Q}) := \sup_{f \in \mathcal{F}} \mathbb{E}_{x \sim \mathbb{P}}[f(x)] - \mathbb{E}_{x \sim \mathbb{Q}}[f(x)], \tag{2}$$

where $\mathcal{F}$ is a class of witness functions $f : \mathcal{X} \to \mathbb{R}$. By choosing different function class $\mathcal{F}$, IPM reduces to many well-known distance metrics between probability distributions. In particular, the Wasserstein-1 distance [Villani, 2008] is defined using the 1-Lipschitz functions $\{f : \|f\|_L \leq 1\}$, where the Lipschitz semi-norm $\|\cdot\|_L$ is defined as $\|f\|_L = \sup_{x \neq y} |f(x) - f(y)|/|x - y|$. Furthermore, total variation is also a kind of IPM and we use $d_{\mathrm{TV}}(\cdot, \cdot)$ to denote it.

# 3 A Lower Bound for Expected Return

In this section, we derive a lower bound for the expected return function in the context of deep MBRL with continuous states and non-linear stochastic dynamics. The lower bound concerns about the expected return, *i.e.*, Eq. (1) and is expressed in the following form [Janner et al., 2019]:

$$\eta[\pi] \geq \hat{\eta}[\pi] - C, \tag{3}$$

where $\hat{\eta}[\pi]$ denotes the expected return of running the policy $\pi$ on a learned dynamics model $\hat{T}(s' \mid s, a)$ and the term $C$ is what we wish to construct. Normally, the dynamics model $\hat{T}$ is learned with experiences $(s, a, s')$ collected by a behavioral policy $\pi_D$ in the real environment dynamics $T$. Typically, in an online MBRL method with iterative policy optimization, the behavioral policy $\pi_D$ represents a collection of past policies. Once we have derived this lower bound, we can naturally design a model-based framework to optimize the RL objective by maximizing the lower bound. Due to page limit, we defer all the proofs to the appendix.

Recall that in MBRL, we have real data $(s, a, s')$ collected in the real dynamics $T$ by the behavioral policy $\pi_D$ and will generate simulated data using the dynamics model $\hat{T}$ with the current policy $\pi$. We begin by showing that for any state $s'$, the discrepancy between its visit distributions in real data and simulated data admits the following decomposition.

**Lemma 3.1.** Assume the initial state distributions of the real dynamics $T$ and the dynamics model $\hat{T}$ are the same. For any state $s'$, assume there exists a witness function class $\mathcal{F}_{s'} = \{f : \mathcal{S} \times \mathcal{A} \to \mathbb{R}\}$ such that $\hat{T}(s' \mid \cdot, \cdot) : \mathcal{S} \times \mathcal{A} \to \mathbb{R}$ is in $\mathcal{F}_{s'}$. Then the following holds:

$$|\nu_T^{\pi_D}(s') - \nu_{\hat{T}}^{\pi}(s')| \leq \gamma d_{\mathcal{F}_{s'}}(\rho_T^{\pi_D}, \rho_{\hat{T}}^{\pi}) + \gamma \mathbb{E}_{(s,a) \sim \rho_T^{\pi_D}} \left| T(s' \mid s, a) - \hat{T}(s' \mid s, a) \right|. \tag{4}$$

Lemma 3.1 states that the discrepancy between two state visit distributions for each state is upper bounded by the dynamics model error for predicting this state and the discrepancy between the two state-action occupancy measures. Intuitively, it means that when both the input state-action distributions and the conditional dynamics distributions are close then the output state distributions will be close as well. Based on this lemma, now we derive the main result that gives a lower bound for the expected return.

**Theorem 3.1.** Let $R := \sup_{s,a} r(s, a) < \infty$, $\mathcal{F} := \cup_{s' \in \mathcal{S}} \mathcal{F}_{s'}$ and define $\epsilon_\pi := 2d_{\mathrm{TV}}(\nu_T^{\pi}, \nu_T^{\pi_D})$. Under the assumption of Lemma 3.1, the expected return $\eta[\pi]$ admits the following bound:

$$\eta[\pi] \geq \hat{\eta}[\pi] - R \cdot \epsilon_\pi - \gamma R \cdot d_{\mathcal{F}}(\rho_T^{\pi_D}, \rho_{\hat{T}}^{\pi}) \cdot \mathrm{Vol}(\mathcal{S}) - \gamma R \cdot \mathbb{E}_{(s,a) \sim \rho_T^{\pi_D}} \sqrt{2D_{\mathrm{KL}}(T(\cdot|s, a) \parallel \hat{T}(\cdot|s, a))}, \tag{5}$$

where $\mathrm{Vol}(\mathcal{S})$ is the volume of state space $\mathcal{S}$.

**Remark** Theorem 3.1 gives a lower bound on the objective in the true environment. In this bound, the last term corresponds to the model estimation error on real data, since the Kullback–Leibler divergence measures the average quality of current model estimation. The second term denotes the divergence between state visit distributions induced by the policy $\pi$ and the behavioral policy $\pi_D$ in the environment, which is an important objective in batch reinforcement learning [Fujimoto et al., 2019] for reliable exploitation of off-policy samples. The third term is the integral probability metric between the $(s, a)$ distributions $\rho_T^{\pi_D}$ and $\rho_{\hat{T}}^{\pi}$, which exactly corresponds to the distribution mismatch problem between model learning and model usage.

---

**Algorithm 1** AMPO

---

1: Initialize policy $\pi_\phi$, dynamics model $\hat{T}_\theta$, environment buffer $\mathcal{D}_e$, model buffer $\mathcal{D}_m$
2: **repeat**
3:     Take an action in the environment using the policy $\pi_\phi$; add the sample$(s, a, s', r)$ to $\mathcal{D}_e$
4:     **if** every $E$ real timesteps are finished **then**
5:         Perform $G_1$ gradient steps to train the model $\hat{T}_\theta$ with samples from $\mathcal{D}_e$
6:         **for** $F$ model rollouts **do**
7:             Sample a state $s$ uniformly from $\mathcal{D}_e$
8:             Use policy $\pi_\phi$ to perform a $k$-step model rollout starting from $s$; add to $\mathcal{D}_m$
9:         **end for**
10:       Perform $G_2$ gradient steps to train the feature extractor with samples $(s, a)$ from both $\mathcal{D}_e$
          and $\mathcal{D}_m$ by the model adaptation loss $\mathcal{L}_{\text{WD}}$
11:     **end if**
12:     Perform $G_3$ gradient steps to train the policy $\pi_\phi$ with samples $(s, a, s', r)$ from $\mathcal{D}_e \cup \mathcal{D}_m$
13: **until** certain number of real samples

---

We would like to maximize the lower bound in Theorem 3.1 jointly over the policy and the dynamics model. In practice, we usually omit model optimization in the first term $\hat{\eta}[\pi]$ for simplicity like in previous work [Luo et al., 2018]. Then optimizing the first term only over the policy and the last term over the model together becomes the standard principle of Dyna-style MBRL approaches. And RL usually encourages the agent to explore, so we won't constrain the policy according to the second term since it violates the rule of exploration, which aims at seeking out novel states. Then the key is to minimize the third term, *i.e.*, the occupancy measure divergence, which is intuitively reasonable since the dynamics model will predict simulated $(s, a)$ samples close to its training data with high accuracy. To optimize this term over the policy, we can use imitation learning methods on the dynamics model, such as GAIL, [Ho and Ermon, 2016] where the real samples are viewed as the expert demonstrations. However, optimizing this term over the policy is unnecessary, which may further reduce the efficiency of the whole training process. For example, one does not need to further optimize the policy using this term but just uses the $\hat{\eta}[\pi]$ term. So in this paper, we mainly focus on how to optimize this occupancy measure matching term over the model.

## 4 AMPO Framework

To optimize the occupancy measure matching term over the model, instead of alleviating the distribution mismatch problem on data level, we tackle it explicitly on feature level from the perspective of unsupervised domain adaptation [Ben-David et al., 2010, Zhao et al., 2019], which aims at generalizing a learner on unlabeled data with labeled data from a different distribution. One promising solution for domain adaptation is to find invariant feature representations by incorporating an additional objective of feature distribution alignment [Ben-David et al., 2007, Ganin et al., 2016]. Inspired by this, we propose to introduce a model adaptation procedure to encourage the dynamics model to learn the features that are invariant to the real state-action data and the simulated one.

Model adaptation can be seamlessly incorporated into existing Dyna-style MBRL methods since it is orthogonal to them, including those by reducing the distribution mismatch problem. In this paper, we adopt MBPO [Janner et al., 2019] as our baseline backbone framework due to its remarkable success in practice. We dub the integrated framework AMPO and detail the algorithm in Algorithm 1.

### 4.1 Preliminary: MBPO Algorithm

**Model Learning**    We use a bootstrapped ensemble of probabilistic dynamics models $\{\hat{T}_\theta^1, ..., \hat{T}_\theta^B\}$ to capture model uncertainty, which was first introduced in [Chua et al., 2018] and has shown to be effective in model learning [Janner et al., 2019, Wang and Ba, 2019]. Here $B$ is the ensemble size and $\theta$ denotes the parameters used in the model ensemble. To be specific, each individual dynamics model $\hat{T}_\theta^i$ is a probabilistic neural network which outputs a Gaussian distribution with diagonal covariance conditioned on the state $s_n$ and the action $a_n$: $\hat{T}_\theta^i(s_{n+1} \mid s_n, a_n) = \mathcal{N}(\mu_\theta^i(s_n, a_n), \Sigma_\theta^i(s_n, a_n))$. The neural network models in the ensemble are initialized differently and trained with different bootstrapped samples selected from the environment buffer $\mathcal{D}_e$, which stores the real data collected

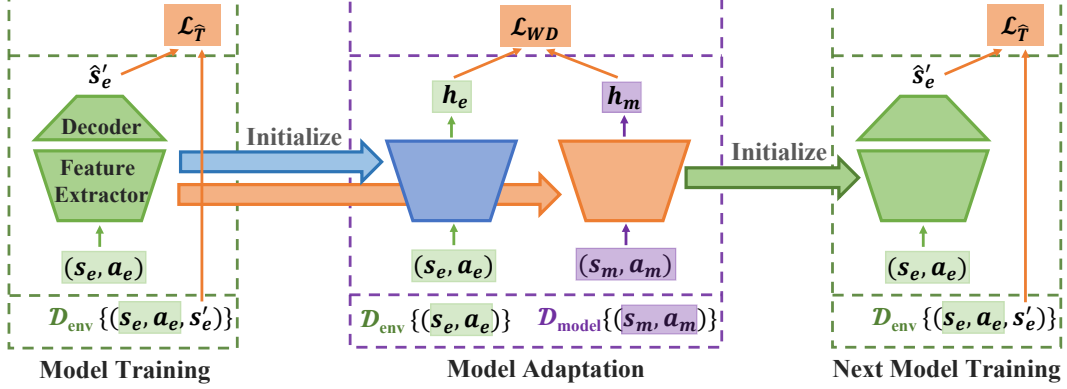

Figure 1: Illustration of model training and model adaptation. At every iteration, the model is learned by maximum likelihood estimation with real data collected from the environment. After the model training, the feature extractor is copied, and then the model adaptation begins where the two separate feature extractors are used for real data and simulated data respectively. After the model adaptation at the current iteration is finished, the feature extractor for the simulated data will be used to initialize the model training at the next iteration.

from the environment. To train each single model, the negative log-likelihood loss is used:

$$\mathcal{L}_{\hat{T}}^i(\theta) = \sum_{n=1}^{N} \left[\mu_\theta^i\left(s_n, a_n\right) - s_{n+1}\right]^\top \Sigma_\theta^{i}{}^{-1}\left(s_n, a_n\right)\left[\mu_\theta^i\left(s_n, a_n\right) - s_{n+1}\right] + \log \det \Sigma_\theta^i\left(s_n, a_n\right).$$

(6)

**Model Usage** The ensemble models are used to generate $k$-length simulated rollouts branched from the states sampled from the environment buffer $\mathcal{D}_e$. In detail, at each step, a model from the ensemble is selected at random to predict the next state and then the simulated data is added to the model buffer $\mathcal{D}_m$. Then a policy is trained on both real and simulated data from two buffers with a certain ratio. We use soft actor-critic (SAC) [Haarnoja et al., 2018] as the policy optimization algorithm, which trains a stochastic policy with entropy regularization in actor-critic architecture by minimizing the expected KL-divergence:

$$\mathcal{L}_\pi(\phi) = \mathbb{E}_s[D_{\mathrm{KL}}(\pi_\phi(\cdot|s) \,\|\, \exp(Q(s_t, \cdot) - V(s)))].$$

(7)

### 4.2 Incorporating Unsupervised Model Adaptation

For convenience, in the following, we only consider one individual dynamics model, and the same procedure could be applied to any other dynamics model in the ensemble. Since the model is implemented by a neural network, we define the first several layers as the feature extractor $f_g$ with corresponding parameters $\theta_g$ and the remaining layers as the decoder $f_d$ with parameters $\theta_d$. Thus we have $\hat{T} = f_d \circ f_g$ and $\theta = \{\theta_g, \theta_d\}$. We propose to add a model adaptation loss over the output of feature extractor, which encourages such a conceptual division as the feature encoder and the decoder. The main idea of model adaptation is to adjust the feature extractor $f_g$ in order to align the two feature distributions of real samples and simulated ones as input, so that the induced feature distributions from real and simulated samples are close in the feature space.

To incorporate unsupervised model adaptation into MBPO, we adopt alternative optimization between model training and model adaptation as illustrated in Figure 1. At every iteration (line 4 to 11 in Algorithm 1), when the dynamics model is trained, we use it to generate simulated rollouts which will then be used for model adaptation and policy optimization. As for the detailed adaptation strategy, instead of directly sharing the parameter weights of the feature extractor between real data and simulated data [Ganin et al., 2016], we choose to adopt the asymmetric feature mapping strategy [Tzeng et al., 2017], which has been shown to outperform the weight-sharing variant in domain adaptation due to more flexible feature mappings. To be specific, the asymmetric feature mapping

strategy unties the shared weights between two domains and learns individual feature extractors for real data and simulated data respectively. Thus in AMPO, after the model adaptation at one iteration is finished, we will use the weight parameters for simulated data to initialize the model training for the next iteration. Through such an alternative optimization between model training and model adaptation, the feature representations learned by the feature extractor will be informative for the decoder to predict real samples, and more importantly it can generalize to the simulated samples.

### 4.3 Model Adaptation via Wasserstein-1 Distance

Specifically, given real samples $(s_e, a_e)$ from the environment buffer $\mathcal{D}_e$ and the simulated samples $(s_m, a_m)$ from the model buffer $\mathcal{D}_m$, the two separate feature extractors map them to feature representations $h_e = f_g^e(s_e, a_e)$ and $h_m = f_g^m(s_m, a_m)$. To achieve model adaptation, we minimize one kind of IPM between the two feature distributions $\mathbb{P}_{h_e}$ and $\mathbb{P}_{h_m}$ according to the lower bound in Theorem 3.1. In this paper, we choose Wasserstein-1 distance as the divergence measure in model adaptation, which is validated to be effective in domain adaptation [Shen et al., 2018]. In the appendix, we also provide a variant that uses Maximum Mean Discrepancy.

Wasserstein-1 distance corresponds to IPM where the witness function satisfies the 1-Lipschitz constraint. To estimate the Wasserstein-1 distance, we use a critic network $f_c$ with parameters $\omega$ as introduced in Wasserstein GAN [Arjovsky et al., 2017]. The critic maps a feature representation to a real number, and then according to Eq. (2) the Wasserstein-1 distance can be estimated by maximizing the following objective function over the critic:

$$\mathcal{L}_{\text{WD}}(\theta_g^e, \theta_g^m, \omega) = \frac{1}{N_e} \sum_{i=1}^{N_e} f_c(h_e^i) - \frac{1}{N_m} \sum_{j=1}^{N_m} f_c(h_m^j). \tag{8}$$

In the meanwhile, the parameterized family of critic functions $\{f_c\}$ should satisfy 1-Lipschitz constraint according to the IPM formulation of Wasserstein-1 distance. In order to properly enforce 1-Lipschitz, we choose the gradient penalty loss [Gulrajani et al., 2017] for the critic

$$\mathcal{L}_{gp}(\omega) = \mathbb{E}_{\mathbb{P}_{\hat{h}}}[(\|\nabla f_c(\hat{h})\|_2 - 1)^2], \tag{9}$$

where $\mathbb{P}_{\hat{h}}$ is the distribution of uniformly distributed linear interpolations of $\mathbb{P}_{h_e}$ and $\mathbb{P}_{h_m}$.

After the critic is trained to approximate the Wasserstein-1 distance, we optimize the feature extractor to minimize the estimated Wasserstein-1 distance to learn features invariant to the real data and simulated data. To sum up, model adaptation though Wasserstein-1 distance can be achieved by solving the following minimax objective

$$\min_{\theta_g^e, \theta_g^m} \max_{\omega} \quad \mathcal{L}_{\text{WD}}(\theta_g^e, \theta_g^m, \omega) - \alpha \cdot \mathcal{L}_{gp}(\omega), \tag{10}$$

where $\theta_g^e$ and $\theta_g^m$ are the parameters of the two feature generators for real data and simulated data respectively, and $\alpha$ is the balancing coefficient. For model adaptation at each iteration, we alternate between training the critic to estimate the Wasserstein-1 distance and training the feature extractor of the dynamics model to learn transferable features.

## 5 Experiments

### 5.1 Comparative Evaluation

**Compared Methods** We compare our method AMPO to other model-free and model-based algorithms. Soft Actor-Critic (SAC) [Haarnoja et al., 2018] is the state-of-the-art model-free off-policy algorithm in terms of sample efficiency and asymptotic performance so we choose SAC for the model-free baseline. For model-based methods, we compare to MBPO [Janner et al., 2019], PETS [Chua et al., 2018] and SLBO [Luo et al., 2018].

**Environments** We evaluate AMPO and other baselines on six MuJoCo continuous control tasks with a maximum horizon of 1000 from OpenAI Gym [Brockman et al., 2016], including InvertedPendulum, Swimmer, Hopper, Walker2d, Ant and HalfCheetah. For the Swimmer environment, we use the modified version introduced by [Langlois et al., 2019] since the original version is quite difficult to solve. For the other five environments, we adopt the same settings as in [Janner et al., 2019].

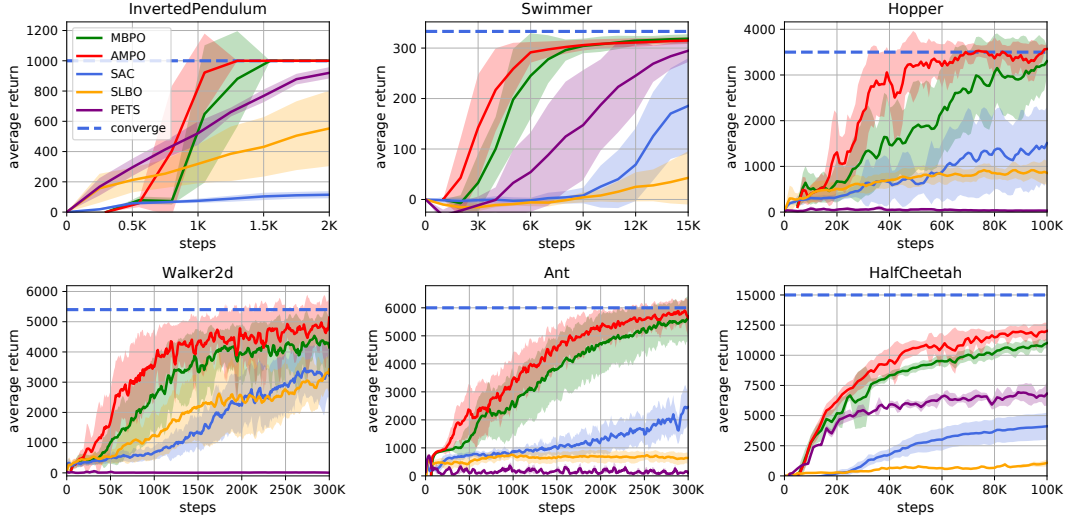

Figure 2: Performance curves of AMPO and other model-based and model-free baselines on six continuous control benchmarking environments. We average the results over five random seeds, where solid curves depict the mean of five trials and shaded areas indicate the standard deviation. The dashed reference lines are the asymptotic performance of Soft Actor-Critic (SAC).

**Implementation Details**  We implement all our experiments using TensorFlow.[2] For MBPO and AMPO, we first apply a random policy to sample a certain number of real data and use them to pre-train the dynamics model. In AMPO, the model adaptation procedure will not be executed any more after a certain number of real samples, which doesn't affect performance. In each adaptation iteration, we train the critic for five steps and then train the feature extractor for one step, and the coefficient $\alpha$ of gradient penalty is set to 10. Every time we train the dynamics model, we randomly sample several real data as a validation set and stop the model training if the model loss does not decrease for five gradient steps, which means we do not choose a specific value for the hyperparameter $G_1$. Other important hyperparameters are chosen by grid search and detailed hyperparameter settings used in AMPO can be found in the appendix.

**Results**  The learning curves of all compared methods are presented in Figure 2. From the comparison, we observe that our approach AMPO is the most sample efficient as they learn faster than all other baselines in all six environments. Furthermore, AMPO is capable of reaching comparable asymptotic performance of the state-of-the-art model-free baseline SAC. Compared with MBPO, our approach achieves better performance in all the environments, which verifies the value of model adaptation. This also indicates that even in the situation with reduced distribution mismatch by using short rollouts, model adaptation still helps.

### 5.2   Model Errors

To better understand how model adaptation affects model learning, we plot in Figure 3(a) the curves of one-step model losses in two environments. By comparison, we observe that both the training and validation losses of dynamics models in AMPO are smaller than that in MBPO throughout the learning process. It shows that by incorporating model adaptation the learned model becomes more accurate. Consequently, the policy optimized based on the improved dynamics model can perform better.

We also investigate the compounding model errors of multi-step forward predictions, which is largely caused by the distribution mismatch problem. The $h$-step compounding error [Nagabandi et al., 2018] is calculated as $\epsilon_h = \frac{1}{h}\sum_{i=1}^{h}\|\hat{s}_i - s_i\|^2$ where $\hat{s}_{i+1} = \hat{T}_\theta(\hat{s}_i, a_i)$ and $\hat{s}_0 = s_0$. From Figure 3(b) we observe that AMPO achieves smaller compounding errors than MBPO, which verifies that AMPO can successfully mitigate the distribution mismatch.

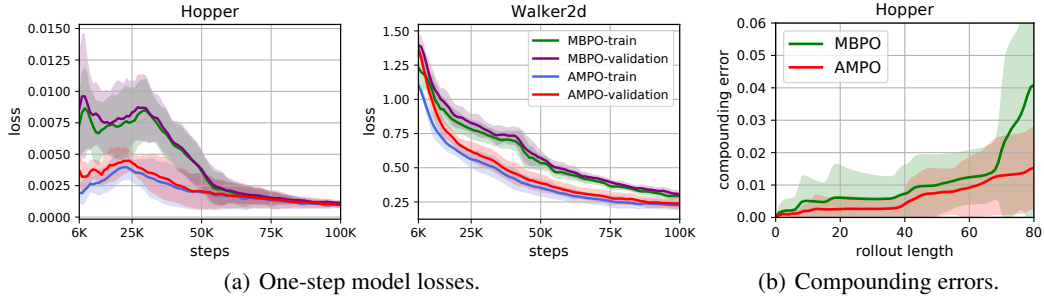

(a) One-step model losses.　　　　　　　　(b) Compounding errors.

Figure 3: (a) The one-step model losses are evaluated on the training and (varying) validation data set from the environment buffer every time the model is trained. (b) Every 5000 environment steps in Hopper, we calculate multi-step compounding errors and then average them.

### 5.3 Wasserstein-1 Distance Visualization

To further investigate the effect of model adaptation, we visualize the estimated Wasserstein-1 distance between the real features and simulated ones. Besides MBPO and AMPO, we additionally analyze the multi-step training loss of SLBO since it also uses the model output as the input of model training, which may help learn invariant features. According to the results shown in Figure 4(a), we find that: i) the vanilla model training in MBPO itself can slowly minimize the Wasserstein-1 distance between feature distributions; ii) the multi-step training loss in SLBO does help learn invariant features but the improvement is limited; iii) the model adaptation loss in AMPO is effective in promoting feature distribution alignment, which is consistent with our initial motivation.

### 5.4 Hyperparameter Studies

In this section, we study the sensitivity of AMPO to important hyperparameters, and the results in Hopper are shown in Figure 4(b). We first conduct experiments with different adaptation iterations $G_2$. We observe that increasing $G_2$ yields better performance up to a certain level while too large $G_2$ degrades the performance, which means that we need to control the trade-off between model training and model adaptation to ensure the representations to be invariant and also discriminative. We then conduct experiments with different rollout length schedules, of which the effectiveness has been shown in MBPO [Janner et al., 2019]. We observe that generating longer rollouts earlier in AMPO improves the performance while it degrades the performance of MBPO a little. It is easy to understand since as discussed in Section 5.2 the learned dynamics model in AMPO obtains better accuracy in approximations and therefore longer rollouts can be performed.

## 6 Related Work

The two important issues in MBRL methods are model learning and model usage. Model learning mainly involves two aspects: (1) function approximator choice like Gaussian process [Deisenroth and Rasmussen, 2011], time-varying linear models [Levine et al., 2016] and neural networks [Nagabandi et al., 2018], and (2) objective design like multi-step L2-norm [Luo et al., 2018], log loss [Chua et al., 2018] and adversarial loss [Wu et al., 2019]. Model usage can be roughly categorized into four groups: (1) improving policies using model-free algorithms like Dyna [Sutton, 1990, Luo et al., 2018, Clavera et al., 2018, Janner et al., 2019], (2) using model rollouts to improve target value estimates for temporal difference (TD) learning [Feinberg et al., 2018, Buckman et al., 2018], (3) searching policies with back-propagation through time by exploiting the model derivatives [Deisenroth and Rasmussen, 2011, Levine et al., 2016], and (4) planning by model predictive control (MPC) [Nagabandi et al., 2018, Chua et al., 2018] without explicit policy. The proposed AMPO framework with model adaptation can be viewed as an innovation in model learning by additionally adopting an adaptation loss function.

In this paper, we mainly focus on the distribution mismatch problem in deep MBRL [Talvitie, 2014], *i.e.*, the state-action occupancy measure used for model learning mismatches the one generated for model usage. Several previous methods have been proposed to reduce the distribution mismatch

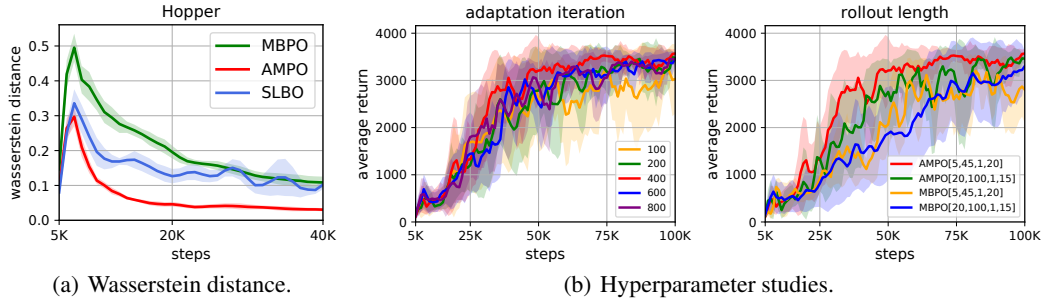

(a) Wasserstein distance.   (b) Hyperparameter studies.

Figure 4: (a) We visualize the Wasserstein-1 distance between the feature distributions. (b) We study the effect of the number of adaptation iterations and the rollout length in AMPO. $[a, b, x, y]$ means the rollout length linearly increases from $x$ to $y$ at the epochs between $a$ and $b$. The rollout schedule $[20,100,1,15]$ is the value used in MBPO and $[5,45,1,20]$ is the schedule we choose for AMPO.

problem. Firstly, it can be reduced by improving model learning, such as using probabilistic model ensembles [Chua et al., 2018], designing multi-step models [Asadi et al., 2019] and adopting a generative adversarial imitation objective [Wu et al., 2019]. Secondly, it can be reduced by designing delicate schemes in model usage, such as using short model-generated rollouts [Janner et al., 2019] and interpolating between rollouts of various lengths [Buckman et al., 2018]. Although these existing methods help alleviate the distribution mismatch, it has not been solved explicitly. On the other hand, the multi-step training loss in SLBO [Luo et al., 2018] and the *self-correct* mechanism [Talvitie, 2014, Talvitie, 2017] can solve this problem. They may also help learn invariant features since the model predicted states were used as the input to train the model in addition to the real data. By comparison, model adaptation directly enforces the distribution alignment constraint to mitigate the problem, and the decoder in AMPO is only trained by real data to guarantee it is unbiased.

Previous theoretical works on MBRL mostly focused on either the tabular MDPs or linear dynamics [Szita and Szepesvári, 2010, Jaksch et al., 2010, Dean et al., 2019, Simchowitz et al., 2018], but not much in the context of continuous state space and non-linear systems. Recently, [Luo et al., 2018] gave theoretical guarantee of monotonic improvement by introducing a reference policy and imposing constraints on policy optimization and model learning related to the reference policy. Then [Janner et al., 2019] also derived a lower bound focusing on branched short rollouts and the algorithm was designed intuitively instead of maximizing the lower bound.

# 7   Conclusion

In this paper, we investigate how to explicitly tackle the distribution mismatch problem in MBRL. We first provide a lower bound to justify the necessity of model adaptation to correct the potential distribution bias in MBRL. We then propose to incorporate unsupervised model adaptation with the intention of aligning the latent feature distributions of real data and simulated data. In this way, the model gives more accurate predictions when generating simulated data, and therefore the follow-up policy optimization performance can be improved. Extensive experiments on continuous control tasks have shown the effectiveness of our work. As a future direction, we plan to integrate additional domain adaptation techniques to further promote distribution alignment. We believe our work takes an important step towards more sample-efficient MBRL.

## Broader Impact

The proposed model adaptation can be incorporated into existing Dyna-style model-based methods, such as MBPO in this paper, to further improve the sample efficiency. This improvement will ease the application of MBRL in practical decision-making problems like robotic control in the future. Despite the potential positive impacts of model adaptation, we should also notice some negative issues. It will cost more to tune real-world MBRL systems with model adaptation to avoid too strong or too weak adaptation, which is usually related to specific environments. We hope our work can provide insights for future improvements in tackling the distribution mismatch problem in MBRL.

## Acknowledgments

The corresponding author Weinan Zhang is supported by "New Generation of AI 2030" Major Project (2018AAA0100900) and National Natural Science Foundation of China (61702327, 61772333, 61632017, 81771937).

## Footnotes

[2]Our code is publicly available at: `https://github.com/RockySJ/ampo`

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
