[Supplementary Material]

# Appendix for: Model-based Policy Optimization with Unsupervised Model Adaptation

## A Omitted Proofs

**Lemma 3.1.** Assume the initial state distributions of the real dynamics $T$ and the dynamics model $\hat{T}$ are the same. For any state $s'$, assume there exists a witness function class $\mathcal{F}_{s'} = \{f : \mathcal{S} \times \mathcal{A} \to \mathbb{R}\}$ such that $\hat{T}(s' \mid \cdot, \cdot) : \mathcal{S} \times \mathcal{A} \to \mathbb{R}$ is in $\mathcal{F}_{s'}$. Then the following holds:

$$|\nu_T^{\pi_D}(s') - \nu_{\hat{T}}^{\pi}(s')| \leq \gamma d_{\mathcal{F}_{s'}}(\rho_T^{\pi_D}, \rho_{\hat{T}}^{\pi}) + \gamma \mathbb{E}_{(s,a) \sim \rho_T^{\pi_D}} \left| T(s' \mid s, a) - \hat{T}(s' \mid s, a) \right|. \quad (4)$$

*Proof.* For the state visit distribution $\nu_{\hat{T}}^{\pi}(s)$, we have

$$\nu_{\hat{T}}^{\pi}(s') = (1 - \gamma)\nu_0(s') + \gamma \int \rho_{\hat{T}}^{\pi}(s, a)\hat{T}(s'|s, a)\,\mathrm{d}s\,\mathrm{d}a \quad (11)$$

where $\nu_0$ denotes the probability of the initial state being the state $s'$. Then we have

$$
\begin{aligned}
&|\nu_T^{\pi_D}(s') - \nu_{\hat{T}}^{\pi}(s')| \\
&= \gamma \left| \int_{s,a} T(s'|s, a)\rho_T^{\pi_D}(s, a) - \hat{T}(s'|s, a)\rho_{\hat{T}}^{\pi}(s, a)\,\mathrm{d}s\,\mathrm{d}a \right| \\
&= \gamma \left| \mathbb{E}_{(s,a) \sim \rho_T^{\pi_D}}[T(s'|s, a)] - \mathbb{E}_{(s,a) \sim \rho_{\hat{T}}^{\pi}}[\hat{T}(s'|s, a)] \right| \\
&\leq \gamma \left| \mathbb{E}_{(s,a) \sim \rho_T^{\pi_D}}[T(s'|s, a) - \hat{T}(s'|s, a)] \right| + \gamma \left| \mathbb{E}_{(s,a) \sim \rho_T^{\pi_D}}[\hat{T}(s'|s, a)] - \mathbb{E}_{(s,a) \sim \rho_{\hat{T}}^{\pi}}[\hat{T}(s'|s, a)] \right| \\
&\leq \gamma \mathbb{E}_{(s,a) \sim \rho_T^{\pi_D}} \left| T(s'|s, a) - \hat{T}(s'|s, a) \right| + \gamma d_{\mathcal{F}_{s'}}(\rho_T^{\pi_D}, \rho_{\hat{T}}^{\pi}),
\end{aligned}
$$
$$\quad (12)$$

which completes the proof. ∎

**Theorem 3.1.** Let $R := \sup_{s,a} r(s, a) < \infty$, $\mathcal{F} := \cup_{s' \in \mathcal{S}} \mathcal{F}_{s'}$ and define $\epsilon_\pi := 2d_{\mathrm{TV}}(\nu_T^{\pi}, \nu_T^{\pi_D})$. Under the assumption of Lemma 3.1, the expected return $\eta[\pi]$ admits the following bound:

$$\eta[\pi] \geq \hat{\eta}[\pi] - R \cdot \epsilon_\pi - \gamma R \cdot d_{\mathcal{F}}(\rho_T^{\pi_D}, \rho_{\hat{T}}^{\pi}) \cdot \mathrm{Vol}(\mathcal{S}) - \gamma R \cdot \mathbb{E}_{(s,a) \sim \rho_T^{\pi_D}} \sqrt{2D_{\mathrm{KL}}(T(\cdot|s, a) \| \hat{T}(\cdot|s, a))}, \quad (5)$$

where $\mathrm{Vol}(\mathcal{S})$ is the volume of state space $\mathcal{S}$.

*Proof.* The return discrepancy is bounded as follows

$$
\begin{aligned}
|\eta(\pi) - \hat{\eta}(\pi)| &= \left| \int_{s,a} \left( \rho_T^{\pi}(s, a) - \rho_{\hat{T}}^{\pi}(s, a) \right) r(s, a)\,\mathrm{d}s\,\mathrm{d}a \right| \\
&= \left| \int_{s,a} \left( \nu_T^{\pi}(s)\pi(a|s) - \nu_{\hat{T}}^{\pi}(s)\pi(a|s) \right) r(s, a)\,\mathrm{d}s\,\mathrm{d}a \right| \\
&\leq R \cdot \int_{s,a} \left| \nu_T^{\pi}(s)\pi(a|s) - \nu_{\hat{T}}^{\pi}(s)\pi(a|s) \right| \mathrm{d}s\,\mathrm{d}a \\
&= R \cdot \int_s \left| \nu_T^{\pi}(s) - \nu_{\hat{T}}^{\pi}(s) \right| \mathrm{d}s \\
&= R \cdot \int_s \left| \nu_T^{\pi_D}(s) - \nu_{\hat{T}}^{\pi}(s) + \nu_T^{\pi}(s) - \nu_T^{\pi_D}(s) \right| \mathrm{d}s \\
&\leq R \cdot \int_s \left| \nu_T^{\pi_D}(s) - \nu_{\hat{T}}^{\pi}(s) \right| \mathrm{d}s + R \cdot \epsilon_\pi
\end{aligned}
$$
$$\quad (13)$$

Replacing the above state $s$ with the notation $s'$, then according to Lemma 3.1, we have

$$|\eta(\pi) - \hat\eta(\pi)|$$

$$\leq R \cdot \epsilon_\pi + \gamma R \cdot \mathbb{E}_{(s,a)\sim\rho_T^{\pi_D}} \int_{s'} \left| T(s'|s,a) - \hat{T}(s'|s,a) \right| \mathrm{d}s' + \gamma R \cdot \int_{s'} d_{\mathcal{F}_{s'}}(\nu_T^{\pi_D}, \nu_{\hat{T}}^\pi) \, \mathrm{d}s'$$

$$\leq R \cdot \epsilon_\pi + \gamma R \cdot \mathbb{E}_{(s,a)\sim\rho_T^{\pi_D}} \int_{s'} \left| T(s'|s,a) - \hat{T}(s'|s,a) \right| \mathrm{d}s' + \gamma R \cdot d_{\mathcal{F}}(\rho_T^{\pi_D}, \rho_{\hat{T}}^\pi) \cdot \mathrm{Vol}(\mathcal{S}) \quad (14)$$

$$= R \cdot \epsilon_\pi + 2\gamma R \cdot \mathbb{E}_{(s,a)\sim\rho_T^{\pi_D}} d_{\mathrm{TV}}(T(\cdot|s,a), \hat{T}(\cdot|s,a)) + \gamma R \cdot d_{\mathcal{F}}(\rho_T^{\pi_D}, \rho_{\hat{T}}^\pi) \cdot \mathrm{Vol}(\mathcal{S})$$

$$\leq R \cdot \epsilon_\pi + \gamma R \cdot \mathbb{E}_{(s,a)\sim\rho_T^{\pi_D}} \sqrt{2D_{\mathrm{KL}}(T(\cdot|s,a), \hat{T}(\cdot|s,a))} + \gamma R \cdot d_{\mathcal{F}}(\rho_T^{\pi_D}, \rho_{\hat{T}}^\pi) \cdot \mathrm{Vol}(\mathcal{S}) \,,$$

where the last inequality holds due to Pinsker's inequality, which completes the proof. ∎

## B  Hyperparameters Settings

Table 1: Hyperparameter settings for AMPO results. $[a, b, x, y]$ denotes a thresholded linear function, *i.e.* at epoch $e$, $f(e) = \min(\max(x + \frac{e-a}{b-a} \cdot (x - y), x), y)$.

| | | Inverted Pendulum | Swimmer | Hopper | Walker2d | Ant | Half Cheetah |
|---|---|---|---|---|---|---|---|
| | network architecture | \multicolumn{6}{c}{MLP with four hidden layers of size 200; feature extractor: four hidden layers; decoder: one output layer} | | | | | |
| | real samples for model pretraining | 300 | 2000 | 5000 | | | |
| | real steps per epoch | 250 | 1000 | | | | |
| | model adaptation batch size | 64 | 256 | | | | |
| $E$ | real steps between model training | 125 | 250 | | | | |
| $F$ | model rollout batch size | 100000 | | | | | |
| $B$ | ensemble size | 7 | | | | | |
| $G_3$ | policy updates per real step | 30 | 20 | | | | 40 |
| $k$ | rollout length | 1 | 1 | [5,45,1,20] | 1 | [10,50,1,20] | [1,30,1,5] |
| $G_2$ | model adaptation updates | 6 | 40 | 400 | 1000 | 3000 | [1,30,100,1000] |
| | model adaptation early stop epoch | 6 | 6 | 40 | 80 | 60 | 30 |

## C  MMD Variant of AMPO

Besides Wasserstein distance, we can use other distribution divergence metrics to align the features. MMD is another instance of IPM when the witness function class is the unit ball in a reproducing kernel Hilbert space (RKHS). Let $k$ be the kernel of the RKHS $\mathcal{H}_k$ of functions on $\mathcal{X}$. Then the squared MMD in $\mathcal{H}_k$ between two feature distributions $\mathbb{P}_{h_e}$ and $\mathbb{P}_{h_m}$ is [Gretton et al., 2012]:

$$\mathrm{MMD}_k^2(\mathbb{P}_{h_e}, \mathbb{P}_{h_m}) := \mathbb{E}_{h_e, h_e'}[k(h_e, h_e')] + \mathbb{E}_{h_m, h_m'}[k(h_m, h_m')] - 2\mathbb{E}_{h_e, h_m}[k(h_e, h_m)], \quad (15)$$

which is a non-parametric measurement based on kernel mappings. In practice, given finite feature samples from distributions $\{h_e^1, \cdots, h_e^{N_e}\} \sim \mathbb{P}_{h_e}$ and $\{h_m^1, \cdots, h_m^{N_m}\} \sim \mathbb{P}_{h_m}$, where $N_e$ and $N_m$ are the number of real samples and simulated ones, one unbiased estimator of $\mathrm{MMD}_k^2(\mathbb{P}_{h_e}, \mathbb{P}_{h_m})$ can

Figure 5: Performance curves of MBPO and MMD variant of AMPO.

be written as follows:

$$\mathcal{L}_{\mathrm{MMD}}(\theta_g) = \frac{1}{N_e(N_e-1)}\sum_{i\neq i'}k(h_e^i, h_e^{i'}) + \frac{1}{N_m(N_m-1)}\sum_{j\neq j'}k(h_m^j, h_m^{j'}) - \frac{2}{N_eN_m}\sum_{i=1}^{N_e}\sum_{j=1}^{N_m}k(h_e^i, h_m^j).$$

(16)

To achieve model adaptation through MMD, we optimize the feature extractor to minimize the above adaptation loss $\mathcal{L}_{\mathrm{MMD}}$ with real $(s, a)$ data and simulated one as input.

When implementing the MMD variant, choosing optimal kernels remains an open problem and we use a linear combination of eight RBF kernels with bandwidths $\{0.001, 0.005, 0.01, 0.05, 0.1, 1, 5, 10\}$. The results on three environments are shown in Figure 5. We observe that using MMD as the distribution divergence measure is also effective in the AMPO framework.

## D   More Experiment Results

### D.1   One-step Model Losses

We show the one-step model losses during the experiments in the other four environments in Figure D.5. We find that the conclusion in Section 5.2 still holds in these four environments. In InvertedPendulum and Swimmer, the standard deviation is a little larger since the number of real samples for pre-training the model is less.

### D.2   Hyperparameter: Policy Updates

In MBRL, since we can generate simulated data using the dynamics model, we can take more gradient updates of policy optimization with the simulated data per environment step to accelerate policy learning. However, too many gradient updates for the policy may cause the current model to be inaccurate for the updated policy. Thus the number of policy gradient updates is a quite important hyperparameter in MBRL. We conduct environments with different policy updates, and show the results in Figure 7(a). We find that when we increase the number of policy updates, the performance of MBPO decreases a little while it doesn't influence AMPO much. It demonstrates that the robustness of AMPO to this hyperparameter.

### D.3   Model Adaptation Early Stopping

According to the model losses in Figure 3(a) and Figure D.5, we find that after certain number of environment steps, the model loss difference between AMPO and MBPO becomes small. So in AMPO we early stop the model adaptation procedure after collecting a certain number of real data, such as 40K in the Hopper environment. We then conduct experiments without early stopping model adaptation and the results are demonstrated in Figure 7(b). We find that keeping adapting the dynamics model throughout the whole learning process does not bring performance improvement. This indicates that model adaptation makes a difference only when the model training data is insufficient. So we set a model adaptation early stopping epoch for each environment (see Table 1 for detail) to improve the computation efficiency.

Figure 6: One-step model losses in other four environments.

(a) Policy updates.　(b) Early stopping.　(c) Computation time.　(d) Adaptation strategy.

Figure 7: More empirical analysis. (a) AMPO-20 means we use AMPO and the number of policy updates is 20. (b) AMPO-NOSTOP denotes the AMPO variant without early stopping the model adaptation procedure. (c) The ratio of computation time of AMPO to that of MBPO. (d) AMPO-SW denotes the AMPO variant of sharing the feature extractor weights for two data distributions. AMPO-ADDA denotes the AMPO variant of fixing the feature extractor of real data.

### D.4　Computation Time

Since AMPO adds the model adaptation procedure based on MBPO, we would like to see its computation time compared with MBPO. We show in Figure 7(c) the computation time ratio of AMPO against MBPO using the same device. We find that in most environments AMPO needs slightly more computation time than MBPO, and the extra overhead is not much. In InvertedPendulum, however, the computational overhead of AMPO is less than MBPO, of which the reason may be that the next model training in AMPO needs less computation after one model adaptation.

### D.5　Adaptation Strategy

In AMPO, we untie the feature extractor weights for two data distributions and learn the two feature extractors simultaneously, which is a variant of the adaptation strategy in Adversarial Discriminative Domain Adaptation (ADDA) [Tzeng et al., 2017]. Differently in ADDA the feature mapping for source domain (*i.e.*real data) is fixed. Another alternative is to share the feature extractor weights between the two data distributions. From the comparison in Figure 7(d), we observe that the performance of these three adaptation strategies differs not much but AMPO performs slightly better.

## E　A Different View of Analysis

In this section, we provide an alternative perspective on the expected return lower bound derivation.

**Lemma E.1.** ([Luo et al., 2018], Lemma 4.3; [Yu et al., 2020], Lemma 4.1) Let $T$ be the real dynamics and $\hat{T}$ be the dynamics model. Let $G_{\hat{T}}^{\pi}(s,a) := \mathbb{E}_{s' \sim \hat{T}(\cdot|s,a)}[V_T^{\pi}(s')] - \mathbb{E}_{s' \sim T(\cdot|s,a)}[V_T^{\pi}(s')]$. Then,

$$\hat{\eta}[\pi] - \eta[\pi] = \gamma \mathbb{E}_{(s,a) \sim \rho_{\hat{T}}^{\pi}}[G_{\hat{T}}^{\pi}(s,a)].$$

Let $\mathcal{F}_1$ be a collection of functions from $\mathcal{S} \times \mathcal{A}$ to $\mathbb{R}$ and $\mathcal{F}_2$ be a collection of functions from $\mathcal{S}$ to $\mathbb{R}$. With Lemma E.1, under the assumption that $G_{\hat{T}}^{\pi}(s,a) \in \mathcal{F}_1$ and $V_T^{\pi}(s') \in \mathcal{F}_2$, we have

$$
\begin{aligned}
\hat{\eta}[\pi] - \eta[\pi] &= \gamma \mathbb{E}_{(s,a)\sim\rho_{\hat{T}}^{\pi}}[G_{\hat{T}}^{\pi}(s,a)] - \gamma \mathbb{E}_{(s,a)\sim\rho_T^{\pi_D}}[G_{\hat{T}}^{\pi}(s,a)] + \gamma \mathbb{E}_{(s,a)\sim\rho_T^{\pi_D}}[G_{\hat{T}}^{\pi}(s,a)] \\
&\leq \gamma \sup_{f\in\mathcal{F}_1} \left| \mathbb{E}_{(s,a)\sim\rho_T^{\pi_D}}[f(s,a)] - \mathbb{E}_{(s,a)\sim\rho_{\hat{T}}^{\pi}}[f(s,a)] \right| \\
&\quad + \gamma \mathbb{E}_{(s,a)\sim\rho_T^{\pi_D}} \left[ \sup_{g\in\mathcal{F}_2} \left| \mathbb{E}_{s'\sim\hat{T}(\cdot|s,a)}[g(s')] - \mathbb{E}_{s'\sim T(\cdot|s,a)}[g(s')] \right| \right] \\
&= \gamma d_{\mathcal{F}_1}(\rho_T^{\pi_D}, \rho_{\hat{T}}^{\pi}) + \gamma \mathbb{E}_{(s,a)\sim\rho_T^{\pi_D}}[d_{\mathcal{F}_2}(\hat{T}(\cdot|s,a), T(\cdot|s,a))].
\end{aligned}
\tag{17}
$$

By rewriting it as a lower bound form, we have

$$
\eta[\pi] \geq \hat{\eta}[\pi] - \gamma d_{\mathcal{F}_1}(\rho_T^{\pi_D}, \rho_{\hat{T}}^{\pi}) - \gamma \mathbb{E}_{(s,a)\sim\rho_T^{\pi_D}}[d_{\mathcal{F}_2}(\hat{T}(\cdot|s,a), T(\cdot|s,a))].
$$

Similarly, if we assume the reward function is bounded, $d_{\mathcal{F}_2}(\hat{T}, T)$ can also be a total variation distance since $\|V_T^{\pi}\|_{\infty}$ is bounded. By comparing this lower bound to the one in Theorem 3.1, it seems this one might be tighter and there is no extra $\epsilon_{\pi}$ term. But we should notice that the assumptions made here are stronger. To be more specific, we assume $G_{\hat{T}}^{\pi}$ satisfies the constraint while in Theorem 3.1 we only assume the model $\hat{T}$ to satisfy the constraint, which is easier to hold.