[Reviews · NeurIPS 2020]

Review 1

Summary and Contributions: The paper addresses the problem of distribution mismatch in model-based RL algorithms between phases of training and using the model. The authors approach this problem by extending the Model-based Policy Optimization (MBPO) from (Janner et al., 2019) with an unsupervised model adaptation procedure. The main contribution of the paper is the derivation of a lower bound for the expected return that involves an integral probability metric (IPM) measuring the discrepancy between the normalised state-action occupancy measures of the behaviour policy under the real dynamics model, and the optimised policy under the estimated dynamics of the environment. This term quantifies the distribution mismatch and is a target to be minimised. To arrive at a practical algorithm, the authors instantiate the IPM of the lower bound with the Wasserstein-1 distance and show how to estimate it with a critic, appropriate loss terms, and an alternating training scheme. Experimental results on six standard OpenAI Gym environments show state-of-the-art results, improving on MBPO in all cases.

Strengths: The target for the main approach of the paper, the unsupervised model adaptation, has strong theoretical grounding in the novel lower bound on the expected return which the authors derive and provide proofs for. These theoretical results are nicely instantiated in the practical algorithm using the Wasserstein-1 distance, leading to impressive SOTA results in the empirical evaluation. I have no doubt that the paper is of high interest for the NeurIPS community.

Weaknesses: One point of weakness that comes to mind is the support of the claim that the algorithm really manages to align the latent distributions of real and simulated data (see Conclusion section). Figure 3 (b) in the paper suggests that the approach does indeed improve alignment over vanilla MBPO, but over long rollouts, there are still significant compounding errors. This is also evident in the statement of the authors that forcing "too much" feature alignment can be hurtful to performance (it should not be if features were really getting aligned better and better). So one should rather speak of approximate alignment or reduced misalignment.

Correctness: Apart from the point mentioned under Weaknesses, it would be good to mention how many trials were used to produce the averaged curves in figures 2, 3, and 4 (I could not find this information).

Clarity: Overall, the paper is very well written and concepts are explained nicely. As a minor note: please check for missing articles throughout the paper.

Relation to Prior Work: The discussion of relevant prior work is nicely structured, very thorough, and delineates the contribution of the paper well.

Reproducibility: Yes

Additional Feedback: One thing that remained unclear to me: in the model adaptation phase (depicted in Fig. 1, middle), are state-action pairs simply sampled randomly from their respective buffers or is there some kind of nearest neighbour selection of real and simulated samples? Should it be random, it would appear rather counterintuitive to minimise the feature distance of arbitrary points in state-action space, rather than making sure that nearby pairs have similar feature activations. Could you please comment on this? Other than that: how important is the division of the model in feature extractor and decoder? Do you have results for a single, monolithic model? Does that lead to less invariant/more biased features? Finally, did you investigate the reasons for the slow learning in the 500 steps on InvertedPendulum (Fig. 2, also holds for vanilla MBPO) compared to PETS? ----- Post-rebuttal update: Thank you for the responses you provided, which helped to clarify the questions I had. However, a point that a fellow reviewer brought up about the somewhat impractical implementation of the occupancy measure matching in the policy optimization leads me to adjust my scores downward slightly. Overall, I still believe this to be a valuable contribution for the community though.


Review 2

Summary and Contributions: This paper looks at the important problem of distribution mismatch in model based reinforcement learning. Opposed to others works (like MBPO), this work does not merely attempt to mitigate the effects of distribution mismatch, but to more actively solve the issue by unsupervised model adaptation. Specifically the proposed method trains a latent space (in one of the hidden layers of the neural network model) such that real samples and simulated samples have matching visitation counts in latent space.

Strengths: See "Summary and contributions". In addition, the hyperparameter ablations are a great addition to convincing the reader of the method's effectiveness.

Weaknesses: The experiments shown in Figure 2 do not outperform MBPO beyond the confidence bounds, but I don't think it's necessary to do this (I believe the novelty of the idea outweighs this). Although would be nice to see where this proposed method strongly outperforms MBPO. Perhaps higher dimensional tasks?

Correctness: All correct (as far as I know).

Clarity: Very clear, well written.

Relation to Prior Work: Prior work discussed in a good amount of detail

Reproducibility: Yes

Additional Feedback: medium points: Cite the Dyna algorithm somewhere? line 46: "better policy optimization in MBRL" maybe something could be said why a Dyna style algorithm is needed at all, opposed to PILCO say which optimizes the policy directly with a model. line 118-119: "usually MBRL allows exploration" but exploration has no meaning if these are not samples from the real world, only samples from the model? line 180-181: "we adopt the asymmetric feature mapping strategy", can you elaborate more why you choose this? (beyond line 184-185). small points: line 41: "For model using," might be missing a word. line 63: "M" is a peculiar choice for transition function? (T, P, or f are common). line 162: "Model Using" --> "Model Usage" line 165: "both buffers", meaning sim and real buffers? Maybe clarify Figure 3: use consistent color coding between Fig 3b and Fig 3c ====== After author response ====== Thank you for the responses. I'll increase by confidence A1) I understand. To clarify I meant "AMPO did not outperform MBPO by much". This isn't meant as a crtitue, only that there may be certain experiments where AMPO really outshines MBPO, which could be interesting to investigate. A2) OK, so maybe "exploration" isn't the correct word choice, it's more about how samples should be drawn. A3) OK great. A4) Great!


Review 3

Summary and Contributions: In this work the authors proposed a model-based RL algorithm that learns a dynamics model with real data sampled from the environment and leverage it to generate simulated data to learn an agent. Different from existing approaches, in AMBPO one explicitly addresses the stationary distribution mismatch between simulated data and real data due to inaccurate model estimation. In particular, the authors first derive a lower bound of the expected return, which naturally motivates a model learning loss that has both a MLE term (similar to standard MBRL approaches) and a novel term to match the simulated and real data distributions. AMBPO then utilizes this new approach to learn a model for policy optimization. Performance of AMBPO is then empirically compared with state-of-the-art methods on benchmark control tasks in terms of sample efficiency and modeling errors.

Strengths: This paper proposes an extension to MBPO on explicitly handling one of the model discrepancy terms in the lower bound (Theorem 3.1) with Wasserstein-1 distribution matching, instead of further upper-bounding it as in MBPO. Motivated by this lower bound the authors then proposed a 2-step new algorithm that firsts train the model with MLE, which is same as MBPO. Then, different from MBPO they later perform an additional adaptation step that aims to matches the discrepancy of the occupation measures (as in the third loss term). This learned model is then used to generate samples for policy optimization, which makes this MBRL procedure sample-efficient. The AMBPO procedure is reasonable and quite well-motivated by the theoretical lower bound in general. Experiments on standard mujoco benchmarks are also quite complete which includes comparisons with state-of-the-art model-based RL methods, and several ablation studies. The results that AMBPO outperforms MBPO on most tasks in terms of performance and compounding error also looks good, and it justifies the tighter lower bound for learning the model with distribution matching.

Weaknesses: In general the first part of the paper that talks about the performance sub-optimality bound is reasonable, and is quite standard from Janner'19. The proofs also look very similar to the original MBPO paper except in this work the term that involves the distribution matching of the occupation measures (which relates to model adaptation) is explicitly dealt with, instead of being further upper-bounded. The contribution to the theoretical part is quite incremental to me. I do have one (relatively major) issue/question on how to use the lower bound in Theorem 3.1 as the loss for policy optimization (so that the AMBPO procedure is solving the \max_\pi \max_M ELBO). While for the model learning part the loss terms stemmed from the RHS are reasonable, it is unclear to me how one can incorporate the third term (that involves distribution matching) in the policy optimization. Unlike Janner'19, where \epsilon_\pi can be viewed as a relative entropy regularizer, here I am not sure how to implement the distribution matching term that also involves \pi in it without further upper-bounding this loss term. Unfortunately this is not explained in the policy optimization part of the paper. One minor point is to understand if a different distribution matching metric (other than Wasserstein-1) can improve the performance (because the assumption in Theorem 3.1 about M may not hold, and it's not easy to verify that in practice). I see the authors mentioned MMD metric, which can be a good potential candidate, it'd be great to see some experiments about that.

Correctness: I did check both the main paper and the appendix, the algorithms, theoretical results and experimental setup look correct.

Clarity: The paper is generally well-written, with algorithms, and experiments quite well-explained. I can also follow the flow of this paper without significant problems.

Relation to Prior Work: Prior work of model-baed reinforcement learning are included in the introduction section, so far I think it covers most references i can think about in this field.

Reproducibility: Yes

Additional Feedback:


Review 4

Summary and Contributions: The paper proposes a model-based RL algorithm, which uses unsupervised model adaptation to minimize the distribution mismatch between real data from the environment and synthetic data from the learned model. They derive a lower bound of the expected return, which suggests to minimize the integral probability metric (IPM) between occupancy measures of the environment and the model. Their algorithm does this by minimizing the IPM between the feature distributions of the real and synthetic data, using domain adaptation techniques. The learned model is then used to generate synthetic data for policy optimization (such as in MBPO). They demonstrate state-of-the-art performance in terms of sample efficiency on 6 continuous control tasks from OpenAI Gym.

Strengths: 1. They provide a novel theoretical bound of the expected return, which justifies the need for model adaptation. 2. They achieve state-of-the-art performance in terms of sample efficiency on the continuous control benchmark environments. 3. Their additional results/visualizations of Figure 3 and 4 are appropriate and further validates their approach, e.g. AMPO achieves lower one-step model losses and lower compounding errors. 4. The model mismatch problem is a very relevant topic in model-based RL.

Weaknesses: The unsupervised model learning procedure alone seems complex, with its own set of hyperparameters.

Correctness: Yes.

Clarity: The explanation of the model adaptation (Section 4.2) needs improvements, especially for readers that are not familiar with unsupervised domain adaptation and asymmetric feature mapping. E.g: - It is unclear what this means: "When the model adaptation is finished, we will use the weight parameters for simulated data to initialize the next model training." What is the "next model training"? Are the weights of the feature extractor frozen at this stage? - Do all the 3 stages of Figure 1 happen at every iteration in Algorithm 1?

Relation to Prior Work: Are there prior works that use domain adaptation techniques to learn models for MBRL?

Reproducibility: Yes

Additional Feedback: Are there any difficulties with the unsupervised model learning procedure? Is it hard to tune the hyperparameters and architecture for this?

[Author Response · NeurIPS 2020]

**Reviewer #1**

**Q1**: ...the claim that the algorithm really manages to align the latent distributions of real and simulated data...

**A1**: The goal of model adaptation is to align the feature distributions, but in the meantime we need to control the trade-off between model training and model adaptation to ensure the representations to be invariant and also discriminative. We will revise the inappropriate statements in the final version.

**Q2**: In the model adaptation phase, are state-action pairs simply sampled randomly from their respective buffers?

**A2**: State-action pairs are randomly sampled since model adaptation is distribution level. Actually this objective doesn't minimize the feature distance of arbitrary $(s, a)$ pairs. Instead it minimizes the distance between feature distributions of two data sets. Constraining nearby $(s, a)$ pairs to have similar features is more related to the Lipschitz continuity of NN.

**Q3**: How important is the division in feature extractor and decoder? Do you have results for a single, monolithic model?

**A3**: AMPO uses the same model architecture as MBPO, which can be regarded as a single monolithic model. The model is *conceptually* divided as feature extractor and decoder and one can regard it as a monolithic model. We propose to add a model adaptation loss over the output of feature extractor, which encourages such a conceptual division.

**Q4**: Did you investigate the reasons for the slow learning in the 500 steps on InvertedPendulum compared to PETS?

**A4**: The reason may be that MPC performs well in the environments with low action dimensions (1 in InvertedPendulum), which also holds in the experiments in the PETS paper, since it is easy to find good actions with limited action proposals.

**Reviewer #2**

**Q1**: The experiments shown in Figure 2 do not outperform MBPO beyond the confidence bounds.

**A1**: AMPO does outperform MBPO according to the results since the shaded area corresponds to standard deviation. For example, if five trials are $[1, 3, 3, 3, 3]$, then the mean is 2.6 and the standard deviation is 0.8. But the maximum value of shaded area as shown in our plots is 2.6+0.8=3.4, which surpasses the maximum value in the five trials, i.e., 3.

**Q2**: Exploration has no meaning if these are not samples from the real world, only samples from the model?

**A2**: We also need exploration when sampling data with the model. Imagine that the model is extremely accurate and we use the policy to sample data only with the model, then exploration is also needed to find a good policy.

**Q3**: Can you elaborate more why you choose the asymmetric feature mapping strategy?

**A3**: Asymmetric feature mapping (unshared weights) has been shown to outperform the weight-sharing variant in domain adaptation, due to more flexible feature mappings. This also holds in our experiments as shown in appendix.

**Q4**: Other medium and small points: ...better policy optimization in MBRL...both buffers...consistent color...etc.

**A4**: Thanks for your suggestions. We will fix these problems accordingly.

**Reviewer #3**

**Q1**: The proofs look very similar to the MBPO paper. The contribution to the theoretical part is quite incremental to me.

**A1**: Our analysis is based on occupancy measure, while MBPO decomposes to each timestep. Moreover, our analysis directly enhances the model training process while MBPO focuses on model usage rather than model training.

**Q2**: ...it is unclear to me how one can incorporate the third term in the policy optimization...

**A2**: We can use imitation learning to optimize this occupancy measure matching term over $\pi$, such as GAIL, where the collected real samples are viewed as the expert and the policy is run on the model to sample data. However, for the alternative training scheme of policy and model, optimizing this term over $\pi$ is not necessary, which may further reduce the efficiency of the whole training process. For example, when the model is sufficiently accurate, one does not need to further optimize $\pi$ using this term but just focuses on the $\hat{\eta}[\pi]$ term. Thus like we omit the model optimization in $\hat{\eta}[\pi]$, we also omit the policy optimization in this term. We are happy to provide more discussions on this in the final version.

**Q3**: Can a different distribution matching metric (other than Wasserstein-1) improve the performance?

**A3**: We have experimented with the MMD variant and observed good results. We will include this in the final version.

**Reviewer #4**

**Q1**: The explanation of model adaptation needs improvements. Are there prior works that use ... for MBRL?

**A1**: We will polish the corresponding writing in the final version. As far as we know, there is no such prior work.

**Q2**: Is it hard to tune the hyperparameters and architecture for the model adaptation?

**A2**: The main hyperparameter needed to tune is adaptation iterations, and it won't cost much to find a good one as shown in Fig.4(b). We use the same model architecture as MBPO, and choose first several layers as feature extractor.

[Meta-Review · NeurIPS 2020]

This paper proposes a method of training and adapting learnt environment models to use for policy optimisation. The authors clearly elucidate on the motivation for the method and the issues of current MBRL and propose a way to adapt their model to minimise distribution mismatch. The experimental results clearly show the benefit in continuous control environments.